# Probabilistic Structural Model Updating with Modal Flexibility Using a Modified Firefly Algorithm

**DOI:** 10.3390/ma15238630

**Published:** 2022-12-03

**Authors:** Zhouquan Feng, Wenzan Wang, Jiren Zhang

**Affiliations:** 1Key Laboratory of Wind and Bridge Engineering of Hunan Province, College of Civil Engineering, Hunan University, Changsha 410082, China; 2Research Institute of Hunan University in Chongqing, Chongqing 401151, China

**Keywords:** structural health monitoring, structural model updating, modified firefly algorithm, modal flexibility, damage detection

## Abstract

Structural model updating is one of the most important steps in structural health monitoring, which can achieve high-precision matching between finite element models and actual engineering structures. In this study, a Bayesian model updating method with modal flexibility was presented, where a modified heuristic optimization algorithm named modified Nelder–Mead firefly algorithm (m-NMFA) was proposed to find the most probable values (MPV) of model parameters for the maximum a posteriori probability (MAP) estimate. The proposed m-NMFA was compared to the original firefly algorithm (FA), the genetic algorithm (GA), and the particle swarm algorithm (PSO) through the numerical illustrative examples of 18 benchmark functions and a twelve-story shear frame model. Then, a six-story shear frame model test was performed to identify the inter-story stiffness of the structure in the original and the damage states, respectively. By comparing the two, the position and extent of damage were accurately found and quantified in a probabilistic manner. In terms of optimization, the proposed m-NMFA was powerful to find the MPVs much faster and more accurately. In the incomplete measurement case, only the m-NMFA achieved target damage identification results. The proposed Bayesian model updating method has the advantages of high precision, fast convergence, and strong robustness in MPV finding and the ability of parameter uncertainty quantification.

## 1. Introduction

Structural model updating is an important part of structural health monitoring. Using monitoring data, structural models can be updated to more realistically represent the actual state of the structures and to obtain mechanical analytical results which are closest to the real state of the structures. In fact, structural model updating is an inverse problem of structural analysis, which can be regarded as an optimization problem. Since the 1960s, the field of structural model updating has developed a variety of methods. However, errors exist widely in experimental testing and modeling which are caused by various factors [1,2]. Shinozuka [3] and Collins et al. [4] introduced an uncertainty concept and used a probability statistical method to evaluate the uncertainty of parameters while updating the model parameters. As a result, structural model updating has had a milestone leap.

Probabilistic model updating methods can be roughly divided into stochastic finite element corrections and finite element corrections based on Bayesian methods. Among them, stochastic finite element correction is a large category, which includes model correction based on the Monte Carlo method and model correction based on the perturbation method. Beck and Katafygiotis et al. [5,6] first introduced the Bayesian method into model updating in 1998 and established a preliminary theoretical framework; they then used the asymptotic approximation method to estimate a posterior probability in the parameter space. Researchers including Beck, Katafygiotis, Yuen, and Feng have conducted a series of in-depth studies on the problems of incomplete measurement, complex likelihood function, and posterior probability that are difficult to solve, and proposed "system modal shape", modal flexibility method, local observation concepts and methods, such as data fusion, to further enrich the relevant theories [7,8,9,10,11,12].

Based on the Bayesian model updating framework, model parameters are embedded in the posteriori probability function. Generally, there are two kinds of methods to solve the maximum a posteriori probability (MAP) estimate, one is the Laplace approximation and the other is random sampling of the posteriori probability distribution function. For a globally identifiable case with a large number of data, we can approximate the posterior probability distribution function as a Gaussian distribution including mean and variance, which mean is the most probable value (MPV), and the obtained MPV is the MAP estimate for the model parameters. Because MPV takes the maximum function value for the posterior probability distribution function, the solution of MPV eventually becomes an optimization problem. After completing the construction of a mathematical model for the optimization problem, the optimization algorithm can be used to find the optimal value of the mathematical model. Due to the limitations of analytical methods, numerical methods have become the most widely used methods for solving optimization problems. They can be mainly divided into two categories. The first type includes traditional optimization algorithms focusing on single-objective optimization problems, such as the Newton Iteration Method (NIM), the Conjugate Direction Method (CDM), etc. The other type includes Heuristic Algorithms (HA) for multi-objective optimization problems. Inspired by bionics, scholars have developed Metaheuristic Algorithms (MA) [13,14]. A meta-heuristic optimization algorithm allows randomness in the optimization process of the algorithm by introducing a mutation factor, thereby reducing the possibility of falling into a local optimum. At present, scholars have proposed many meta-heuristic optimization algorithms, such as the Genetic Algorithm (GA) [15], the Ant Colony Optimization (ACO) [16], the Firefly Algorithm (FA) [17], etc. 

The firefly algorithm was first proposed by Yang in 2007. It is a global optimization algorithm inspired by the glowing behavior of fireflies in nature. Since the firefly algorithm was proposed, due to its simple parameters, fast convergence speed, high precision, as well as its own automatic population segmentation ability and nonlinear individual attraction mechanism, it has received extensive attention from the academic community. However, like other swarm intelligence optimization algorithms, this algorithm is prone to problems such as premature maturity, falling into local optimum, and insufficient ability to solve multi-dimensional optimization problems. Therefore, it needs to be appropriately improved when solving actual complex optimization problems. 

Based on the original FA algorithm, this paper makes several modifications and introduces the Nelder–Mead Algorithm (NMA) for local search and proposes a modified Nelder–Mead firefly algorithm. Three improvements are proposed: modification in the control parameters, introduction of a boundary constrain mechanism, and introduction of a diversity threshold. Eighteen benchmark functions and a 12-story shear frame model are used to test the performance of the m-NMFA and to provide support for the comparison of the original FA, the GA, and the PSO to the m-NMFA. A six-story steel shear frame was designed to perform a vibration experiment for verifying the effectiveness of the m-NMFA in Bayesian structural model updating.

## 2. Model Updating Formulation

### 2.1. Modal Flexibility

The stiffness matrix ***K*** and flexibility matrix ***C*** of a linear dynamical system with *N* degrees of freedom can be written as follows [18]:(1)$$K=M\Phi \Lambda \Phi^{T}M=M\left(\sum_{i=1}^{N}\omega_{i}^{2}\phi_{i}\phi_{i}^{T}\right)M.$$
(2)$$C=\Phi \Lambda^{-1}\Phi^{T}=\overset{N}{\underset{i=1}{\sum }}\frac{1}{\omega_{i}^{2}}\phi_{i}\phi_{i}^{T}$$
where ***M*** and ***K*** are the mass matrix and the stiffness matrix, respectively; $\Phi =[\phi_{1},\;\phi_{2},\ldots ,\;\phi_{n}]$ is the mass-normalized mode shape matrix; and ***Λ*** is the diagonal eigenvalue with squared natural circular modal frequencies. 

In practice, it is always the case that the spatial arrangement of sensors attached to the structure is generally limited and only a limited number of DOFs can be observed, for which only incomplete spatial mode shapes can be gained. It can be seen from Equation (2) that the greater the modal frequency is, the smaller the contribution of one mode to the flexibility matrix ***C*** will be, which is on the contrary to that of Equation (1) and is more practical in solving vibration-related problems. Therefore, the flexibility matrix ***C*** can be established approximately with only low-frequency modes of interest. Therefore, the Equation (2) yield as follows [19]:(3)$$C^{m}\approx C_{t}^{m}=\overset{N_{m}}{\underset{i=1}{\sum }}\frac{1}{\omega_{i}^{2}}\phi_{i}^{m}{\left(\phi_{i}^{m}\right)}^{T}$$
where $C^{m}$ denotes the flexibility submatrix with respect to the measured DOFs; $C_{t}^{m}$ denotes the truncated flexibility submatrix; *N_m_* denotes the number of selected low-frequency modes (frequency from low to high and no interval in this research); and $\phi_{i}^{m}\in R^{N_{o}\times 1}{\;(i=1,\;2,\;\ldots ,\;N}_{m})$ denotes the incomplete mode shapes. Needless to say, $C_{t}^{m}$ is supposed to be as close as possible to $C^{m}$; if all the DOFs of a structure are well observed and all the mode shapes are gained, $C_{t}^{m}$ will be exactly equal to $C^{m}$.

Mass-normalization is a key issue to build up the flexibility matrix. If the measurement is incomplete, the dimension of modal parameters, such as mode shapes, would be less than that of the mass matrix ***M***. To handle this problem, the SEREP method [20] is introduced to convert ***M*** to reduced mass matrix $M^{m}$. Then, the incomplete mode shape matrixes are normalized as follows:(4)$${\left(\Phi^{m}\right)}^{T}M^{m}\Phi^{m}=I$$

### 2.2. Model Reduction

System equivalent reduction and expansion process (SEREP) is an eigenvalue-based reduction technique used to approximate the real structural state vectors with mode shape measured from the DOFs of the user’s interest [20].

In the method of SEREP, the first step is to separate the mass and stiffness matrices as follows: (5)$$M^{P}=\left[\begin{array}{cc} M_{mm} & M_{mu}\\ M_{um} & M_{uu} \end{array}\right],\;\;K^{P}=\left[\begin{array}{cc} K_{mm} & K_{mu}\\ K_{um} & K_{uu} \end{array}\right].$$
where *m* and *u* denote the measured and unmeasured DOFs. Then, the SEREP transformation matrix can be written as follows:(6)$$T=\left[\begin{array}{c} \Phi^{m}\\ \Phi^{u} \end{array}\right]{\left(\Phi^{m}\right)}^{*}.$$
where $\Phi^{m}$ denotes the retained mode shape matrix with $N_{o}$ (the number of observed DOFs) rows and $N_{m}$ columns, and ${\left(\Phi^{m}\right)}^{*\;}$ represents the generalized inverse of matrix $\Phi^{m}$. The process of converting $\Phi^{m}$ to ${\left(\Phi^{m}\right)}^{*\;}$ is shown in [11,20,21]. Ultimately, the reduced mass and stiffness matrices can be written as follows: (7)$$M^{m}=T^{T}M^{P}T,\;K^{m}=T^{T}K^{P}T.$$

### 2.3. Bayesian Model Updating Based on Measured Modal Flexibility Data

Bayesian method for finite element model updating is one of the most commonly used probabilistic methods for structural model updating, and it has been confirmed to be efficient and useful [22,23]. In this section, modal flexibility data are used to establish a posterior-PDF-based objective function. 

As the modal flexibility matrix is symmetrical, we can use the lower triangular matrix of the flexibility matrix $C^{l}$ to establish the objective function. Let *f* denote the modal flexibility data, and the matrix $C^{l}$ is vectorized based on the following equation: (8)$$f=\left(c_{11},c_{21},\ldots ,c_{N_{o}1},\ldots ,c_{ii},c_{\left(i+1\right)i},\ldots ,c_{N_{o}i},\ldots ,c_{N_{o}N_{o}}\right)$$
where c denotes the element in the matrix $C^{l}$.

It is supposed that the test is repeated ***N_t_*** times and ***D*** denotes the experimental data. Then, the matrix ***D*** can be written as follows:(9)$$D=\left\{{\hat{f}}_{1},{\hat{f}}_{2},\ldots ,{\hat{f}}_{N_{t}}\right\}$$

As the modal can be described by the probability distributions of both the unknown parameters and the prediction error [5], the measured data can be written as follows: (10)$${\hat{f}}_{t}=f\left(\theta \right)+e_{t}$$
where $f\left(\theta \right)$ denotes the prediction of the analytical model and is defined as the vector for the given values of the model parameters θ. The term $e_{t}$ denotes the prediction error and is assumed to follow the maximum entropy principle [17]. That is to say, $e_{t}$ yields a multi-dimensional Gaussian distribution with zero mean and covariance matrix ***Σ*** [5]. The covariance matrix is computed based on the modal flexibility vectors as follows:(11)$$\Sigma =\frac{1}{N_{t}-1}\overset{N_{t}}{\underset{t=1}{\sum }}\left({\hat{f}}_{t}-\bar{f}\right)\;{\left({\hat{f}}_{t}-\bar{f}\right)}^{T}$$
where $\bar{f}$ contains the average value of $N_{t}$ experimental modal flexibility vectors. It can be expressed using the following equation: (12)$$\bar{f}=\frac{1}{N_{t}}\overset{N_{t}}{\underset{t=1}{\sum }}{\hat{f}}_{t}$$

Then, the likelihood function of the *t*-th set of modal flexibility vector data can be expressed as follows:(13)$$p\left({\hat{f}}_{t}|\theta ,M\right)=C_{1}exp\left(-\frac{1}{2}{\left({\hat{f}}_{t}-f\left(\theta \right)\right)}^{T}\Sigma^{-1}({\hat{f}}_{t}-f\left(\theta \right))\right)$$
where $C_{1}$ is a constant. Assuming that the $N_{t}$ sets of the modal flexibility vector datas are mutually independent, the likelihood function of *D* can be written as follows:(14)$$p\left(D|\theta ,M\right)=C_{2}exp\left(-\frac{1}{2}\overset{N_{t}}{\underset{t=1}{\sum }}{\left({\hat{f}}_{t}-f\left(\theta \right)\right)}^{T}\Sigma^{-1}({\hat{f}}_{t}-f\left(\theta \right))\right)$$
where $C_{2}$ is also a constant like $C_{1}$. As the prior PDF p(θ) follows a non-informative uniform distribution, the posterior PDF p(θ|D,M) can be written in a similar form as the likelihood function in Equation (14) Given that the observed DOFs are limited, the SEREP method will be utilized here and then the measured mode shapes are mass-normalized with respect to the reduced model mass $M^{m}$. It is easy to see that $M^{m}$ depends on θ only and so does the posterior PDF p(θ|D,M). On this basis, the most probable stiffness parameter vector $\hat{\theta }$ can be gained by maximizing the posterior PDF value or minimizing its negative value. Here, the objective function is defined as a truncated part of Equation (14) and is simplified with the following logarithm:(15)$$J\left(\theta \right)=-\ln p(\theta |D,M)$$
where *J*(θ) denotes the objective function. Substitute Equation (14) into Equation (15), the function is expressed as follows:(16)$$J\left(\theta \right)=\frac{1}{2}\overset{N_{t}}{\underset{t=1}{\sum }}{\left({\hat{f}}_{t}-f\left(\theta \right)\right)}^{T}\Sigma^{-1}({\hat{f}}_{t}-f\left(\theta \right))$$

Till now, the objective function has been established and the model updating problem has been transformed as an optimization problem. When the most probable stiffness parameter vector $\hat{\theta }$ is found during the process of minimizing J(θ), the actual stiffness matrix of a structure is ensured naturally. 

The posterior PDF can be well approximated using a set of multi-dimensional Gaussian distribution with $\hat{\theta }$ and $\Sigma_{\theta }$ as its mean value and covariance matrix.
(17)$$\Sigma_{\theta }={\left[H\left(\hat{\theta }\right)\right]}^{-1}={\left[\nabla J(\hat{\theta })\nabla^{T}\nabla J(\hat{\theta })\nabla^{T}\right]}^{-1}$$
where ***H***($\hat{\theta }$) is the Hessian matrix of objective function.

In order to search for the best-fit vector $\hat{\theta }$ exactly and reliably, a modified FA algorithm is proposed, which is introduced in the next section and utilized in the numerical and experimental examples in Section 4 and Section 5.

## 3. Firefly Algorithm

### 3.1. Original Firefly Algorithm

Firefly Algorithm (FA) was first proposed by Xin-She Yang at Cambridge University in 2007 [24] and has become a widely used global optimization algorithm. In the original FA, it is not gender difference but the intensity of light *I*, which decreases with the increase in the square of Cartesian distance between two fireflies *r*^2^, that distinguishes fireflies. Fireflies that have greater *I* values are regarded as fitting the solution to the objective function better, and they are supposed to attract the ones with smaller values to them. Given that light will gradually weaken in a medium as distance increases, the attractiveness parameter *β*, which denotes the attraction between any two fireflies, is introduced in this algorithm. Similar to the intensity of light, the greater the distance is, the smaller the attractiveness parameter *β* will be, and the attraction between two fireflies will become weaker. Furthermore, the light absorption parameter *γ* is introduced to denote the ability of a medium to absorb light, which is proposed to simulate the natural environment that fireflies are in.

The behavior of firefly is idealized for algorithmic use, and three rules are introduced as follows [24]:All the fireflies are unisex so they can attract each other without the influence of gender.The attractiveness is proportional to the brightness, and they both decrease as their distance increases.The brightness of a firefly is determined by the landscape of the objective function.

In a given medium, light intensity I decreases in a Gaussian form as follows:(18)$$I\left(r\right)=I_{0}e^{-\gamma r^{2}}$$
where ${\;I}_{0}$ represents the intensity at *r* = 0, and *r* denotes the distance between any two fireflies. It can be seen that light intensity decreases exponentially and the influence of the absorption coefficient γ on light intensity is defined in the exponential term of the function mentioned above. However, light intensity is flexible in both definition and form since it is allowed to be the fitness value of a certain objective function to a practical problem [17]. 

The attractiveness *β* of a firefly is proportional to the light intensity *I* and is also defined in an exponential form as follows:(19)$$\beta =\beta_{0}e^{-\gamma r^{2}}$$
where $\beta_{0}$ is the attractiveness value at *r* = 0. 

The distance *r* between any adjacent fireflies *x_i_* and *x_j_* is expressed as follows:(20)$$r=||x_{i}-x_{j}||=\sqrt[{2}]{\overset{n}{\underset{k=1}{\sum }}{\left(x_{i,k}-x_{j,k}\right)}^{2}}$$
where *x_i_*_,_*_k_* is the *k*-th component of the spatial coordinate *x_i_* of *i*-th firefly, and *n* denotes the dimension of the objective function or problems to be optimized.

As mentioned above, fireflies with a greater value of light intensity *I* is supposed to attract the ones with a smaller value of *I*. Assume that the *i*-th firefly is weaker than the *j*-th, then the movement of firefly *i* to firefly *j* is described as follows:(21)$$x_{i}=x_{i}+\beta_{0}e^{-\gamma r^{2}}\left(x_{j}-x_{i}\right)+\alpha \epsilon_{i}$$
where the second term is the attraction part, and the third term is the randomization part with step parameter α. *ϵ_i_* is drawn from different distributions, such as Gaussian distribution, uniform distribution, and levy flight. In many cases, a uniform distribution is utilized as the simplest form and, thus, *ϵ_i_* becomes a vector of random numbers uniformly generated in [17,25,26,27]. An example of a levy flight distribution is shown in [24].

The firefly algorithm is an efficient algorithm, and Algorithm 1’s pseudocode is shown as follows:
**Algorithm 1** the firefly algorithmInitialize the parameters(α, β, γ, n)Initialize randomly a population of n fireflies   Evaluate the fitness of the initial population at *x_i_* by       objective function        **While** (k < MaxGen) do           **For** i =1:n              **For** j = 1:n                 **If** Firefly j is better than i                       Firefly i moves towards j                 **End if**              **End for**
              Evaluate the new solution            **End for**           Rank and update the best solution found so far           Update iteration counter k;           Update α         **End while**

### 3.2. Modified Nelder–Mead Firefly Algorithm

The modified Nelder–Mead firefly algorithm(m-NMFA) is generally a hybrid version combining the advantages of the local optimization algorithm, the Nelder–Mead algorithm (NMA), and the global optimization algorithm. In order to strengthen the superiority of the hybrid algorithm, three improvements are proposed: modification in the parameters *α* and *β*, introduction of a boundary constrain mechanism, and introduction of a diversity threshold. 

#### 3.2.1. Modification in Parameters α and β

Multiple studies have proved that the parameter *α* and the attractiveness parameter *β* are of great influence in both convergence speed and accuracy of results [24,24,25,26,27,28,29,30,31,32,33].

Yang in 2008 [17] used a fixed-step parameter *α* = 0.2 to solve a four-peak function and then pointed out that a decreasing *α* could improve the convergence of the FA. Two examples are given as follows: (22)$$\alpha =\alpha_{\infty }+\left(\alpha_{0}-\alpha_{\infty }\right)e^{-t}$$
and
(23)$$\alpha =\alpha_{0}\theta^{t}$$
where $\alpha_{0}$ is the initial step parameter and $\alpha_{\infty }$ is the final value. *θ* ∈[0,1] is the randomness reduction constant. *α* iterates as the iteration counter t increases. It can be seen that the step parameter *α* is limited between $\alpha_{0}$ and $\alpha_{\infty }$ in Formula (22) and gradually approaches $\alpha_{\infty }$. Similarly, *α* gradually approaches zero in Equation (23). Moreover, in Equation (23), the *θ* and $\alpha_{0}$ values have great influence in the decay speed of step parameter *α* in the process of iteration, which will produce a unknown effect—positive or negative—on the solution result. In previous studies, different *θ* values are taken to cope with different problems. Yang and He [34] pointed out that *θ* could take values from 0.95 to 0.97, Manoharan [30] and Rizk-Allah [35] took *θ* as 0.9, and Wang [31] took *θ* as 0.7 and $\alpha_{0}$ as 0.25. Moreover, a noticeable issue has been highlighted that the scaling parameter *S_d_* should be introduced in consideration of the actual scales of the problem of interest [36].
(24)$$S_{d}=u_{d}-l_{d}$$
where *d* denotes the *d*-th dimension unknown variable, and *u_d_* and *l_d_* are the upper and lower limit values. Therefore, $\epsilon_{i}$ in Equation (21) should be replaced by $\epsilon_{i}S$, where *S*={*S*_d_, d=1,2,…n}. 

Beside the strategies mentioned above, various versions of the step parameter iteration formula have been proposed in the last decade. Brajević [27] proposed a novel strategy that iterates α with the maximum number of interaction and the initial step parameter $\alpha_{0}$ and it is written as follows:(25)$$\alpha_{\kappa }^{\left(t\right)}=\alpha_{\kappa }^{\left(t-1\right)}{\left(\frac{1}{90,000}\right)}^{\frac{2}{MaxGen}}$$
where κ is the identifier of each firefly and takes the dimension of the problem as its maximum value; *MaxGen* denotes the maximum number of interaction; and *t* denotes the current number of iterations. The step parameter decreases more quickly and will be much smaller than that in Equation (23). Ignoring the distinction among all the fireflies, Equation (25) can be expressed by another form as follows:(26)$$\alpha \left(t\right)=\alpha_{0}\Theta^{\frac{\rho t}{MaxGen}}$$
where $\Theta ={\left(1/90,000\right)}^{2}\approx 1.234\times 10^{-10}$, and *ρ* is defined as the reference coefficient and its value is 1. Previous tests have shown that the results drawn from an *α* that is of a slightly smaller value during the previous iterations (take 50 iterations for example) tend to outperform the results drawn from a greater *α* value in convergence speed. Previous tests have also shown that the convergence speed of the algorithm increases but the accuracy decreases gradually if ρ is taken to be an integer constant value ranging from 1 to 3. That is to say, the decrease rate of *α* in a later course of iteration should be well controlled to have a good balance between exploration and exploitation to the maximum. So, let ρ be expressed in a nonlinear form using the following equation:(27)$$\rho =1+2{\left(\frac{MaxGen-t}{MaxGen}\right)}^{2}$$

Figure 1 shows the decreasing process of the reference coefficient. 

Apart from the modification on the step parameter, extensive efforts have been made to modify the attraction parameter *β* and the light absorption parameter *γ*. If *β*_0_ and *γ* are taken to be constants, the distance between any two adjacent fireflies will approach zero gradually. As the iteration number increases, it will ideally follow the equation below: (28)$$\underset{t\to \infty }{lim}\beta =\underset{t\to \infty }{lim}\beta_{0}e^{-\gamma r^{2}}=\beta_{0}$$

This ensures that the attraction within a firefly colony will always be a contributing factor. In practice, however, it would be a tough problem in high-dimensional problems if the distance between any two firefly individuals is too large at the very beginning of the iteration progress and the attraction is too weak to get the individuals closer. As a consequence, the algorithm stagnates. Moreover, the higher the dimension of a project is, the more easily this phenomenon will occur. 

Selvarasu [33] proposed a modified β formula:(29)$$\beta =\beta_{min}+\left(\beta_{max}-\beta_{min}\right)e^{-\gamma r^{2}}$$
where *β_min_* and *β_max_* are the lower and upper boundaries of the attraction parameter β. This strategy prevents the attraction from being excessively large or small. Herein, this paper utilizes Equation (29) to improve the performance of the FA in solving high-dimensional problems. 

#### 3.2.2. Boundary Constrain Mechanism

Similar to other swarm intelligence methods, such as the Bee Colony Algorithm (BCA), the Genetic Algorithm (GA), and the Particle Swarm Optimization (PSO), the FA will be ill-conditioned if the variables of firefly individuals exceed the feasible region. Yang introduced a boundary constrain mechanism that forces the outer variables to the border, and the formula is expressed as follows:(30)$$x_{i,j}=\left\{\begin{array}{c} l_{j,}\\ u_{j,} \end{array}\right.\begin{array}{c} x_{i,j}\leq l_{j}\\ x_{i,j}\geq u_{j} \end{array}$$
where $x_{i,j}$ denotes the *i*-th firefly individual and the *j*-th variable, and $l_{j}$ and $u_{j}$ are the lower and upper boundaries of the *j*-th variable. On the one hand, this mechanism ensures that all the variables of one firefly are within the feasible range and, moreover, prevents the algorithm from falling into the local optimum or leading to a completely wrong solution. On the other hand, it reduces the diversity of the firefly population to some extent. This mechanism has been utilized by Ivona Brajević [37] and Rajan [38].

The optimal solution of a multi-dimensional problem is determined by all the variables. It is possible that a newly generated firefly individual would deviate from the optimal solution with only several variables being updated. Therefore, a new boundary constraint mechanism is proposed. If any one of the variables overflows, a new firefly individual will be regenerated randomly, Algorithm 2 is:
**Algorithm 2** boundary constrain mechanism**For** i = 1:n**     For** j = 1:D         **If**
$x_{i,j}{<l}_{j}$ or $x_{i,j}{>u}_{j}$
            **Then**
$x_{i,j}{=l}_{j}{+\mathrm{rand}\times (u}_{j}{-l}_{j})$
         **End if**     **End for**x_i_ = x_i_ + *F**(x_best_ − x_i_)**End for**
where rand∈[0,1] denotes a random factor that is used to generate a new variable randomly, and F∈[0,0.5] is another random factor that leads the new firefly towards the current global best solution *x_best_*. Both *rand* and F follow a uniform distribution. 

#### 3.2.3. Hybrid of Nelder–Mead Algorithm and the Diverse Threshold

The Nelder–Mead Algorithm is a local optimal algorithm that adapts itself to the local landscape and contracts on to the final minimum. This method for finding the local minimum is widely used and it can be realized expediently by using the built-in function “fminsearch” in MATLAB software. The way of combining the NMA and the FA is to start the NMA in the iterative process of the FA in order to dig out a better solution nearby the current global best solution found up to *t*-th (t ≤ MaxGen) iteration. 

However, the start point of the NMA in the iterative process needs to be well weighted because when to enable the NMA will greatly affect the solution accuracy and convergence speed. If the NMA starts too early, the current best solution will be far from reliable and the NMA will run meaninglessly, which will result in inefficiency. Meanwhile, if it starts too late, the contribution of the solution searched by the NMA to the algorithm will be weakened. Therefore, the concept of diversity threshold is introduced. When the diversity of a firefly population reaches a certain threshold, the NMA is enabled to balance the contradiction mentioned above. 

Let $x_{k,\mathrm{best}}$ and $x_{k,\mathrm{worst}}$ denote the best and worst firefly individuals of the single *k*-th iteration, and $x_{g,\mathrm{best}}$ and $x_{g,\mathrm{worst}}$ denote the current global best and worst individuals found up till the *k*-th iteration. The diversity threshold is calculated as follows:(31)$$\xi =\frac{x_{k,best}-x_{k,worst}}{x_{g,best}-x_{g,worst}}$$
(32)$${T=e}^{\xi }-1$$

Then, the NMA is enabled when $e^{\xi }-1<T_{0},\;\mathrm{where}\;$*T*_0_∈[0, e^−1^]. The FA would be capable of finding a relatively stable solution close to the target when $e^{\xi }-1$ is from 0.0001 to 0.01, that is to say, it is feasible to take *T*_0_ from 10^−3^ to 10^−5^, and then the step parameter *α* would be small enough to correspond with the NMA. 

## 4. Numerical Illustrative Examples 

In order to demonstrate the superiority of the newly modified algorithm, 18 benchmark functions and a 12-story shear frame model are used to test the performance of the m-NMFA and to provide support for the comparison of the original FA, the GA [39] and the PSO [40] to the modified algorithm.

### 4.1. Benchmark Work of m-NMFA

Table 1 presents 18 well-known benchmark functions [41,42], and the order number (column 1), the function name (column 2), the formula (column 3), the feasible region (column 4), the minimum value (column 5), and the dimension (column 6) of each function are given successively. These functions can be generally divided into two categories: multimodal functions (*f_1_*–*f_9_*) and unimodal functions (*f_10_*–*f_18_*). The multimodal functions are used to prove the ability of the proposed algorithm to escape the local optimal solution and seek the global optimal solution, and the unimodal functions are used to illustrate the precision of the results searched out by the algorithm.

In this test, all the population size and the number of chromosomes are taken to be 30. The maximum number of iterations is taken to be *MaxGen* = 1000. For the FA algorithms, the initial value of the step parameter is taken to be 0.5 in the m-NMFA and 0.25 in the original FA; the attractiveness parameters *β*_min_ and *β*_max_ take the values of 0.2 and 1.0, respectively; and the absorption coefficient *γ* is taken to be 1.0. For the GA, the test uses the built-in GA optimization algorithm of MATLAB R2017a Win64, and most of the input parameters take the default values of the built-in toolbox, except for the number of chromosomes, the maximum number of iterations, and *FunctionTolerance* which is defined as a stop criterion. When the average relative change in the best fitness function value over *MaxGen* generations is less than or equal to *FunctionTolerance*, the algorithm stops. Given that it demands the same iteration number for all the algorithms, the criterion value should be significantly small and, therefore, takes the value of 10^−100^ in the test. For the PSO, the learning factors take the value of $c_{1}{=c}_{2}=2$, the maximum velocity is taken to be 1.0, and the inertia weighting factor is in a linear form: (33)$$\omega_{t}=\omega_{2}-t\times \frac{\omega_{2}-\omega_{1}}{MaxGen}$$
where $\omega_{i}$ is the inertia weighting factor of the *i*-th iteration, and $\omega_{1}$ and $\omega_{2}$ are the lower and upper limit values of *ω*. If the velocity or spatial vector of a particle in one dimension reaches or exceeds the boundary, all dimensions of the particle will be reset to the boundary value. All the functions are run independently using each of the optimization method for 100 times to avoid stochastic discrepancy in MATLAB R2017a Win64, Windows 10 system.

Table A1 shows the results of the m-NMFA under different diversity threshold values, and additionally, the second to fifth columns show the results of the improved algorithm without the NMA. In each case, the mean value, the standard deviation (SD), and the maximum (Max) and minimum (Min) values of the results of each 100 runs are listed. It can be seen from the results that, compared to the case without the NMA, the other three cases are generally better, which proves that the introduction of the NMA can effectively improve the accuracy and stability of the firefly algorithm. In addition, although taking *T*_0_ = 10^−1^ can improve the optimization performance of the algorithm to a certain extent, the results are not good enough. When *T* = 10^−3^–10^−5^, the accuracy and stability of the optimization algorithm are better than the results of *T* = 10^−1^, which shows that the start point of the local optimization algorithm in the global optimization algorithm has a significant impact on the optimization result. This test also proves that a diversity threshold value in the range of ${T=10}^{-3}{\sim 10}^{-5}$ is relatively appropriate. In subsequent tests and studies, the m-NMFA will adopt the value of *T* = 10^−5^.

Table A2 shows the mean values, the standard deviations, and the maximum and minimum values of every 100 runs. It can be seen that the modified algorithm performs generally the same as the other three algorithms in *f*_1_ and *f*_2_, and it is slightly inferior to the GA in *f*_7_ and *f*_8._ As for the minimum values of *f*_10_ and *f*_14_, m-NMFA performs slightly worse than PSO and GA. Apart from the cases mentioned above, the m-NMFA significantly outperforms the FA, the GA, and the PSO in both precision and dispersion in most cases. It is worth mentioning that the ability of the m-NMFA to search for the minimum values of multimodal functions is well-illustrated in the tests of *f*_7_ and *f*_8_ for its distinguished minimum values out of the other values. 

Figure 2 shows the convergence diagrams of the 18 functions operated by the FA, the GA, the PSO, and the m-NMFA. The value of the ordinate is the mean value after 100 operations. It can be seen that the m-NMFA converges faster in most cases and has a strong high-precision solution searching ability.

### 4.2. Numerical Simulation of Shear Frame

A 12-story steel shear frame shown in Figure 3 is used to test the performance of the m-NMFA. The frame is assumed to be fixed on a solid ground and is of 1.8 m in height. The masses are uniformly distributed on each floor and are totally supported by six full-length rectangular columns with an identical cross-section of 6 mm in thickness and 25 mm in width. The mass of each floor is taken to be *m_e_* = 75 kg, and the inter-story stiffness is calculated to be *k_e_* = 1.512 × 10^5^ N/m. Hence, the nominal undamaged structure mass and stiffness matrices are calculated as follows: (34)$$M_{u}={\left[\begin{array}{cccc} m_{e} & & & \\ & m_{e} & & \\ & & \ddots & \\ & & & m_{e} \end{array}\right]}_{12\times 12}$$
(35)$$K_{u}={\left[\begin{array}{ccccc} 2k_{e} & -k_{e} & & & \\ -k_{e} & 2k_{e} & -k_{e} & & \\ & -k_{e} & \ddots & \ddots & \\ & & \ddots & \ddots & -k_{e}\\ & & & -k_{e} & 2k_{e} \end{array}\right]}_{12\times 12}$$

It is assumed to have a stiffness reduction of overall 20%, 40%, and 20% on the 5th, 6th, and 7th inter-story columns, which is denoted by the artificially settled damage vector $\theta_{\mathrm{ith}}\left(i=5,6,7\right)$. Gaussian white noises with mean values of zero are considered for the modal data, and the degrees of which are taken to be 1% for modal frequencies and 3% for modal shapes. Complete and incomplete measurements are considered. In the complete measurement case, all the first 12 modal parameters can be well measured. In the incomplete case, it is assumed that only the first eight modal parameters and the special displacement information of the 1st, 2nd, 4th, 6th, 8th, 10th, 11th, and 12th stories can be obtained. In order to eliminate accidental errors, the tests are executed independently for $N_{t}=100$ times and 100 sets of modal parameters are used. The simulation progress is operated using the FA, the GA, the PSO, and the m-NMFA separately and the algorithm parameters are kept the same as those in Section 4.1. 

In this example, the standard deviation (SD) and the coefficient of variances (CV) are calculated for each parameter, which are listed in Table 2 and Table 3. Figure 4 shows the results of the damage vector *θ* for both cases. It can be seen that, in the complete measurement case, all the algorithms are able to identify the position and the degree of the damage ideally. However, in the incomplete measurement case, only the m-NMFA can achieve the goal of identifying the damage of a structure.

## 5. Experimental Illustrative Example

A six-story steel shear frame is designed to perform a vibration experiment, in which the modal parameters are identified by FDD to form a modal flexibility matrix. Then, the m-NMFA is utilized to search out the minimum value of the modal-flexibility-based objective function. Damage identification problem is introduced to illustrate the effectiveness of the proposed method.

The structure is fixed on the ground by foundation bolts. The length, width, and thickness of the interlayer plates are 450 mm, 450 mm, and 10 mm, respectively. The thickness center spacing of each plate is 250 mm. The interlayer stiffness is provided by the main columns fixed at the four vertices of the plates and additional columns at the midpoint of the side edges. An angle steel is welded with a steel plate in the center of it. An external excitation of random force and random position is applied to the structure by hammering excitation. The displacement response of each layer is measured using an IL-300 laser displacement meter. The measuring point is set at the center of the angle steel baffle. The sampling frequency of the meter is 200 Hz. The masses from m1 to m6 are 17.151 kg, 17.123 kg, 17.215 kg, 17.169 kg, 17.193 kg, and 16.731 kg, respectively, and the nominal interlaminar stiffness is 81920n/m. Before the test, a lumped mass reference finite element model is established according to the measured mass of each layer and the calculated nominal stiffness.

The displacement responses of the steel shear frames in the damaged and undamaged states are tested. In the damaged state, the additional columns of the fifth floor are missing, of which the nominal damage coefficient is −20%. In both tests, the responses of the 1st, 2nd, 4th, 5th, and 6th stories are measured, and the measurement repeats 50 times for each test and lasts for 5 min each. Figure 5 and Figure 6 show the time domain information of the undamaged and damaged structures. 

The frequencies of the first six modes of the frame structure are shown in Table 4, which take the average values of 50 measurements. The mode shapes shown in Figure 7 are normalized with respect to the reduced mass matrix $M^{m}$.

In the Bayesian probabilistic approach, the probability distributions of the model updating parameters of the damaged and undamaged structures are needed to calculate the cumulative probability distribution of the damage parameters in the damaged state. The distributions of the damage parameters can be obtained using the following formula:(36)$$P_{n}\left(d\right)=P\left(\left({1+\theta }_{n}^{d})<\left(1-d\right)({1+\theta }_{n}^{\mathrm{ud}}\right)\right)\approx \Phi \left(\frac{\left(1-d\right)\left({1+\hat{\theta }}_{n}^{\mathrm{ud}}\right)-\left({1+\hat{\theta }}_{n}^{d}\right)}{\sqrt{{\left(1-d\right)}^{2}{\left(\sigma_{n}^{\mathrm{ud}}\right)}^{2}+{\left(\sigma_{n}^{d}\right)}^{2}}}\right)$$
where the damage extent *d* is used to describe the value of the stiffness reduction parameter; $\theta_{n}^{d}$ and $\theta_{n}^{\mathrm{ud}}$ are the model updating parameters of the damaged and undamaged structures; and ${\hat{\theta }}_{n}^{d}$, ${\hat{\theta }}_{n}^{\mathrm{ud}}$, ${\hat{\sigma }}_{n}^{d}$ and ${\hat{\sigma }}_{n}^{\mathrm{ud}}$ are the mean values and standard deviations of the model updating parameters in Gaussian probability distribution. The value of *θ_n_* can be obtained by choosing the mean value of each distribution in the probability density function diagram. The standard deviations of the *n*-th damage parameters can be calculated using the following formula:(37)$$\sigma_{n}=\frac{\sum_{i}^{\infty }{{(d}_{n,i}{-\theta }_{n})}^{2}{y(x}_{n,i})\Delta d}{\sum_{j}^{\infty }{y(d}_{n,j})\Delta d}$$
where $\Delta {d=d}_{n,v+1}{-d}_{n,v},v\in \{i,\;j|i\in N^{*},\;j\in N^{*}\}$, and y(d) is the Gaussian probability density function of *d*.

The model updating parameters of the finite element model of the undamaged structure are identified using the m-NMFA, which are shown in Table 5. The model updating parameters of the damaged structure are shown in Table 6.

According to the identification results shown in Table 5 and Table 6, the cumulative probability distribution of the structural damage parameter of each floor is calculated using Formula 35. The cumulative probability distribution is plotted in Figure 8. The probability density function is plotted in Figure 9. Figure 10 shows the mean value of each distribution in the probability density function directly.

## 6. Conclusions

Based on the modal flexibility matrix, this paper uses the Bayesian probability method to establish the objective function for the model updating problem, which is solved using the proposed m-NMFA. The results show that this method can accurately fix the damage position and determine the damage extent.

Compared to the other meta-heuristic optimization algorithms, the proposed m-NMFA has a stronger optimization ability in solving benchmark functions and the finite element model updating objective functions, which shows a higher applicability in the practical problems of model updating. The reason why the m-NMFA has more superior optimization performance is mainly due to the following three points:In the research on the random step size formula, it is found that the smaller the random step size is at the beginning of the iteration, the faster the convergence speed of the algorithm will be. However, as the random step size gradually approaches zero at the end of the iteration, its value should be large enough to keep the algorithm with sufficient exploration ability.In the research on the value of diversity threshold, a fraction is used to quantify the diversity of the population to enable the NMA algorithm to start at an appropriate time. The diversity threshold is quantized into a value between (0, e^−1^] through exponential form. When the diversity threshold is taken to be between 10^−3^~10^−5^, the algorithm can obtain a more accurate optimal solution.The selection of the iterative formula of the attraction parameter has great impact on the solving ability of the FA in solving the multi-dimensional optimization problems, The selection of the iterative formula of the attraction parameter has a great impact on the ability of the FA in solving multi-dimensional optimization problems, which would lead to stagnation and non-convergence if an improper formula were selected. This paper avoids such problem by selecting an appropriate formula for the coefficient of attraction.

In the research of model updating based on modal flexibility and Bayesian method, the modal flexibility theory, the system equivalent reduction and expansion process, and the basic theory of Bayesian method are introduced. The objective function established based on these theories provides a relatively complete theoretical support for solving the model updating problems of limited measuring points and number of modes, so that people can obtain reliable results from limited structural information. Bayesian method allows people to empirically select the prior probability distribution of the modified parameters so as to exert influence on the identification results, which provides theoretical and practical convenience for solving the problems of model updating. Through the numerical simulation of a 12-story shear frame and the experimental test of a 6-story steel shear frame, the reliability of the Bayesian probability method in solving model updating and structural damage identification is verified.

## Figures and Tables

**Figure 1 materials-15-08630-f001:**
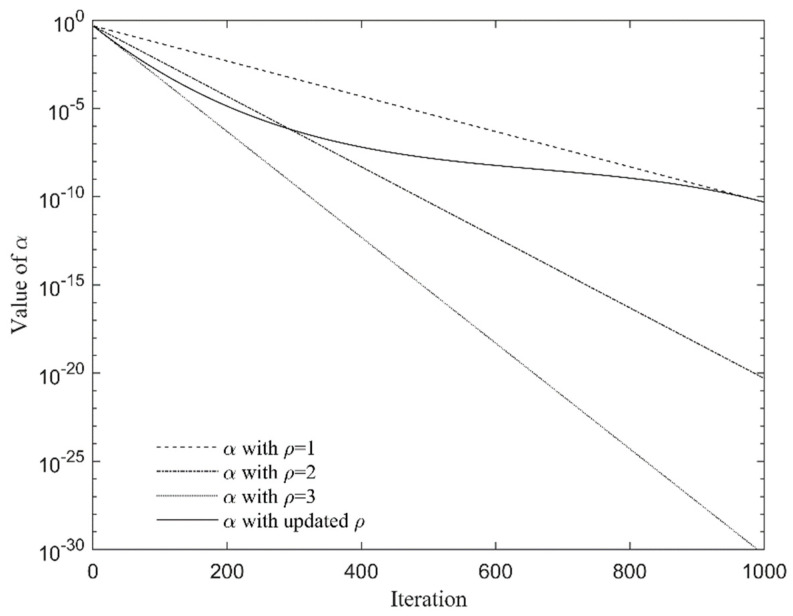
Iteration of the reference coefficient (Θ = 1.0 × 10^−10^, $\alpha_{0}=0.5$).

**Figure 2 materials-15-08630-f002:**
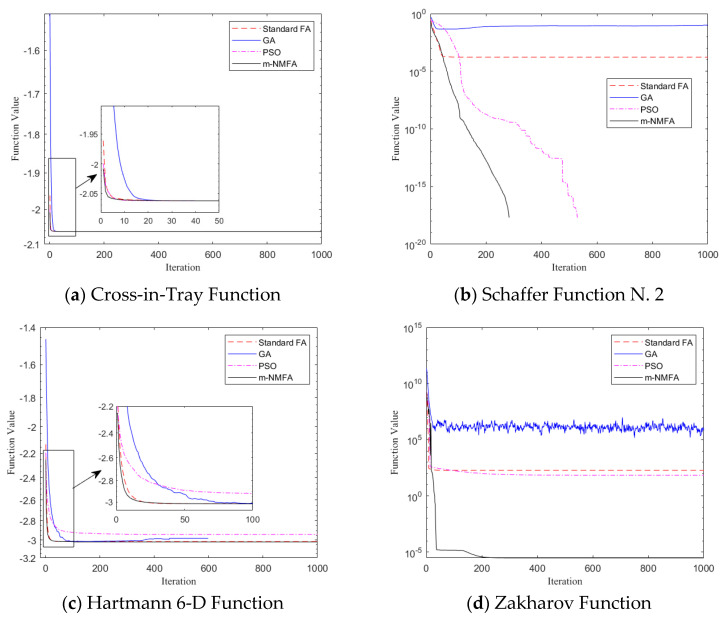
Convergence diagram of four optimization algorithms.

**Figure 3 materials-15-08630-f003:**
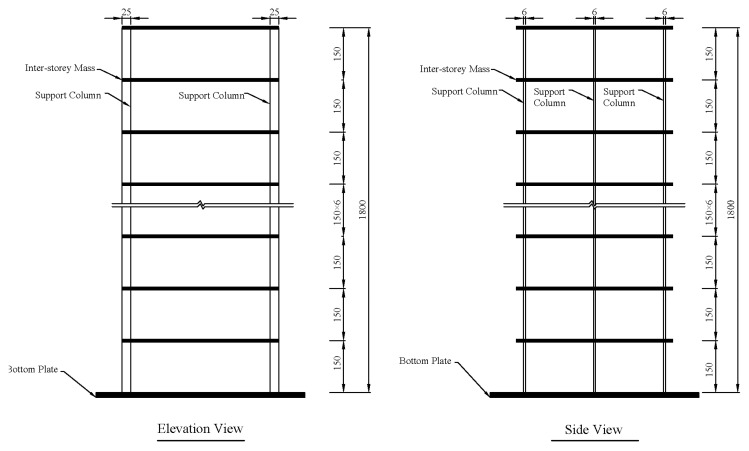
Schematic diagram of a 12-story steel shear frame (Unit: mm).

**Figure 4 materials-15-08630-f004:**
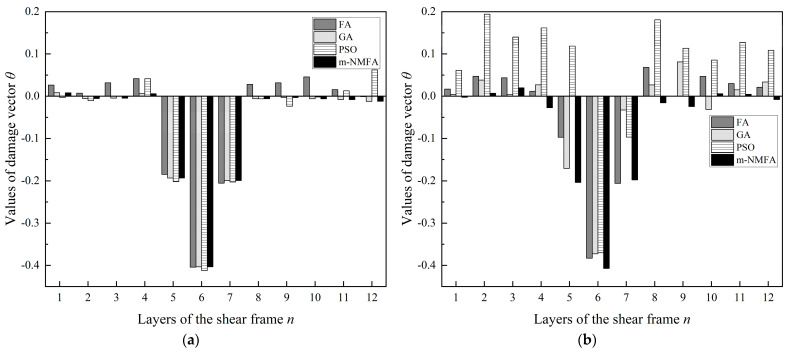
The results of damage vector *θ.* (**a**) Complete measurement case. (**b**) Incomplete measurement case.

**Figure 5 materials-15-08630-f005:**
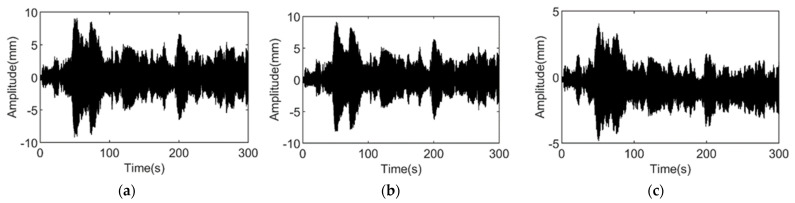
Time domain information of undamaged structure vibration. (**a**) The 1st story; (**b**) the 2nd story; and (**c**) the 4th story.

**Figure 6 materials-15-08630-f006:**
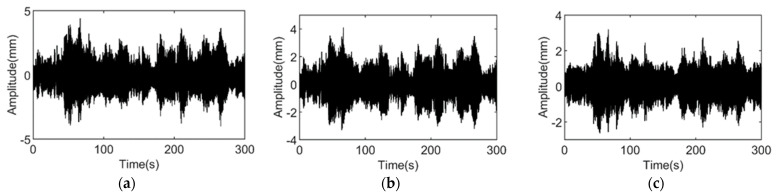
Time domain information of damaged structure vibration. (**a**) The 1st story; (**b**) the 2nd story; and (**c**) the 4th story.

**Figure 7 materials-15-08630-f007:**
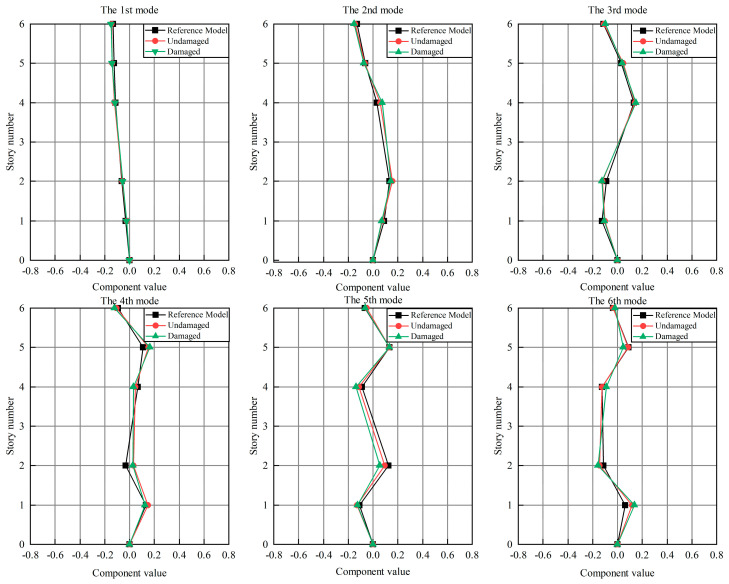
The first six mode shapes of steel shear frames identified by FDD.

**Figure 8 materials-15-08630-f008:**
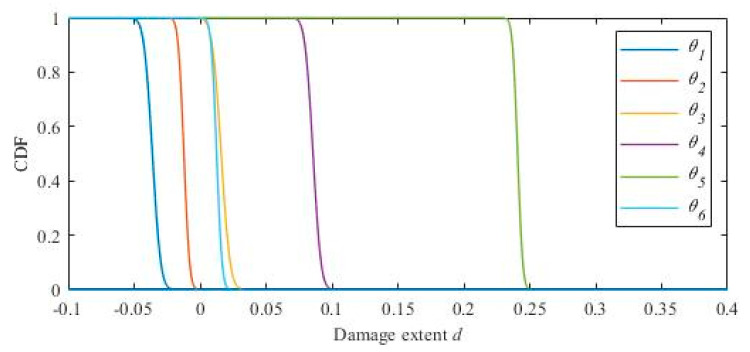
The cumulative probability function of the damage parameters.

**Figure 9 materials-15-08630-f009:**
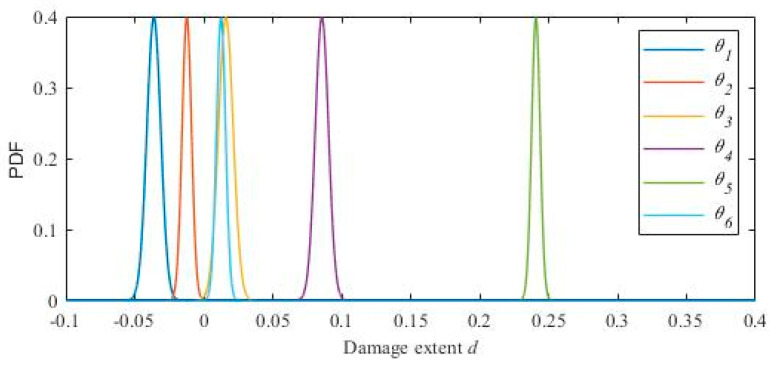
The probability density function of the damage parameters.

**Figure 10 materials-15-08630-f010:**
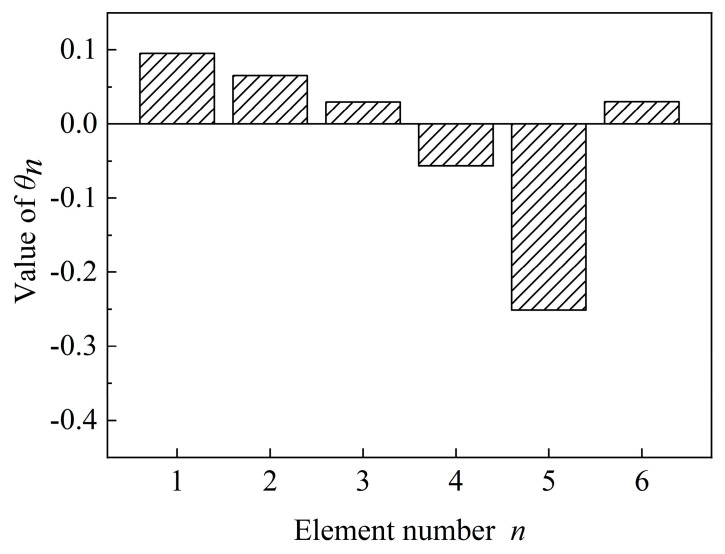
The damage parameter of the shear frame.

**Table 1 materials-15-08630-t001:** Benchmark functions.

No.	Function Names	Formulations	Limits	Min	D
$f_{1}$	Cross-in-Tray Function	$f_{1}\left(x\right)=-0.0001{\left(\left|{\sin(x}_{1}{)\sin(x}_{2})\exp\left(\left|100-\frac{\sqrt{x_{1}^{2}{+x}_{2}^{2}}}{\pi }\right|\right)\right|+1\right)}^{0.1}$	*x_i_*∈[−10,10]	−2.0626	2
$f_{2}$	Schaffer N2 Function	$f_{2}\left(x\right)=0.5+\frac{{\left(\sin\left(x_{1}^{2}{-x}_{2}^{2}\right)\right)}^{2}-0.5}{{\left[1+0.001\left(x_{1}^{2}{+x}_{2}^{2}\right)\right]}^{2}}$	*x_i_*∈[−100,100]	0	2
$f_{3}$	Hartmann 6-D Function	$f_{3}\left(x\right)=-\overset{4}{\underset{i}{\sum }}\alpha_{i}\exp\left(-\overset{6}{\underset{j=1}{\sum }}A_{\mathrm{ij}}{\left(x_{j}{-P}_{\mathrm{ij}}\right)}^{2}\right)$, where $\alpha ={\left(1.0,1.2,\;3.0,3.2\right)}^{T}$ *A*$=\left(\begin{array}{cccccc} 10 & 3 & 17 & 3.5 & 1.7 & 8\\ 0.05 & 10 & 17 & 0.1 & 8 & 14\\ 3 & 3.5 & 1.7 & 10 & 17 & 8\\ 17 & 8 & 0.05 & 10 & 0.1 & 14 \end{array}\right)$ *P*${=10}^{-4}\left(\begin{array}{cccccc} 1312 & 1696 & 5569 & 124 & 8283 & 5886\\ 2329 & 4135 & 8307 & 3736 & 1004 & 9991\\ 2348 & 1451 & 3522 & 2883 & 3047 & 6650\\ 4047 & 8828 & 8732 & 5743 & 1091 & 381 \end{array}\right)$	*x_i_*∈(0,1)	−3.3224	6
$f_{4}$	Zakharov Function	$f_{4}\left(x\right)=\overset{D}{\underset{i=1}{\sum }}x_{i}^{2}+{\left(\overset{D}{\underset{i=1}{\sum }}0{.5\mathrm{ix}}_{i}\right)}^{2}+{\left(\overset{D}{\underset{i=1}{\sum }}0{.5\mathrm{ix}}_{i}\right)}^{4}$	*x_i_*∈[−5,10]	0	30
$f_{5}$	Alpine Function	$f_{5}\left(x\right)=\overset{D}{\underset{i=1}{\sum }}\left|x_{i}\sin\left(x_{i}\right)+0{.1x}_{i}\right|$	*x_i_*∈[−10,10]	0	30
$f_{6}$	Griewank Function	$f_{6}\left(x\right)=\overset{n}{\underset{i=1}{\sum }}\frac{x_{i}^{2}}{4000}-\overset{n}{\underset{i=1}{\prod }}\cos\left(\frac{x_{i}}{\sqrt{i}}\right)+1$	*x_i_*∈[−600,600]	0	30
$f_{7}$	Penalized 1 Function	$f_{7}\left(x\right)=\frac{\pi }{D}\left\{{10\sin}^{2}\left({\pi y}_{1}\right)+\overset{D-1}{\underset{i=1}{\sum }}{\left(y_{i}-1\right)}^{2}\left[{1+10\sin}^{2}\left({\pi y}_{i+1}\right)\right]+{\left(y_{D}-1\right)}^{2}\right\}+\overset{D}{\underset{i=1}{\sum }}u\left(x_{i},10,100,4\right),$ where $y_{i}=1+\frac{1}{4}\left(x_{i}+1\right),\;u\left(x_{i},\;a,\;k,\;m\right)=\left\{\begin{array}{c} k{\left(x_{i}-a\right)}^{m},\\ 0,\\ k{\left({-x}_{i}-a\right)}^{m}, \end{array}\right.\begin{array}{c} x_{i}>a\\ -a\leq x_{i}\leq a\\ x_{i}<-a \end{array}$	*x_i_*∈[−50,50]	0	30
$f_{8}$	Penalized 2 Function	$f_{8}\left(x\right)=\frac{1}{10}\left\{\sin^{2}\left({3\pi x}_{1}\right)+\overset{D-1}{\underset{i=1}{\sum }}{\left(x_{i}-1\right)}^{2}\left[{1+\sin}^{2}\left({3\pi x}_{i+1}\right)\right]+{\left(x_{D}-1\right)}^{2}\left[{1+\sin}^{2}\left({2\pi x}_{D}\right)\right]\right\}+\overset{D}{\underset{i=1}{\sum }}u\left(x_{i},5,100,4\right),$	*x_i_*∈[−50,50]	0	30
$f_{9}$	Ackley Function	$f_{9}\left(x\right)=-20\exp\left(-0.2\sqrt{\frac{1}{n}\overset{n}{\underset{i=1}{\sum }}\cos\left({2\pi x}_{i}\right)}\right)+20+e$	*x_i_*∈[−32.768,32.768]	0	30
$f_{10}$	Sum of Different Powers Function	$f_{10}\left(x\right)=\overset{D}{\underset{i=1}{\sum }}{|x}_{i}{|}^{i+1}$	*x_i_*∈[−1,1]	0	30
$f_{11}$	Sphere Function	$f_{11}\left(x\right)=\overset{D}{\underset{i=1}{\sum }}x_{i}^{2}$	*x_i_*∈[−5.12,5.12]	0	30
$f_{12}$	Sum Squares Function	$f_{12}\left(x\right)=\overset{D}{\underset{i=1}{\sum }}{\mathrm{ix}}_{i}^{2}$	*x_i_*∈[−5.12,5.12]	0	30
$f_{13}$	Rosenbrock Function	$f_{13}\left(x\right)=\overset{D-1}{\underset{i=1}{\sum }}(100{\left(x_{i+1}{-x}_{i}^{2}\right)}^{2}{+(x}_{i}{-1)}^{2})$	*x_i_*∈[−2.048,2.048]	0	30
$f_{14}$	Dixon-Price Function	$f_{14}\left(x\right)={\left(x_{1}-1\right)}^{2}+\overset{D}{\underset{i=2}{\sum }}i{\left({2x}_{i}^{2}{-x}_{i-1}\right)}^{2}$	*x_i_*∈[−10,10]	0	30
$f_{15}$	Rotated Hyper-Ellipsoid Function	$f_{15}\left(x\right)=\overset{D}{\underset{i=1}{\sum }}\overset{i}{\underset{j=1}{\sum }}x_{j}^{2}$	*x_i_*∈[−65.536, 65.536]	0	30
$f_{16}$	Perm Function 0, d, β	$f_{16}\left(x\right)=\overset{D}{\underset{i=1}{\sum }}{\left(\overset{D}{\underset{j=1}{\sum }}\left(j+10\right)\left(x_{j}^{i}-\frac{1}{j^{i}}\right)\right)}^{2}$	*x_i_*∈[−30,30]	0	30
$f_{17}$	Schwefel 1.2 Function	$f_{17}\left(x\right)=\overset{D}{\underset{i=1}{\sum }}{\left(\overset{i}{\underset{j=1}{\sum }}x_{j}\right)}^{2}$	*x_i_*∈[−100,100]	0	30
$f_{18}$	Schwef 2.22 Function	$f_{18}\left(x\right)=\overset{D}{\underset{i=1}{\sum }}\left|x_{i}\right|+\overset{D}{\underset{i=1}{\prod }}\left|x_{i}\right|$	*x_i_*∈[−100,100]	0	30

**Table 2 materials-15-08630-t002:** The results of shear frame: complete measurement case.

No.	*θ* _a_	FA			GA			PSO			m-NMFA		
		*θ*	SD	CV	*θ*	SD	CV	*θ*	SD	CV	*θ*	SD	CV
1	0	0.026	0.0012	0.0452	0.0083	0.0011	0.1374	−0.0030	0.0020	0.6884	0.0081	0.0011	0.1408
2	0	0.007	0.0016	0.2196	−0.0053	0.0015	0.2858	−0.0104	0.0029	0.2772	−0.0054	0.0015	0.2785
3	0	0.032	0.0014	0.0452	−0.0044	0.0013	0.2932	0.0004	0.0042	9.3279	−0.0047	0.0013	0.2729
4	0	0.042	0.0015	0.0350	0.0061	0.0013	0.2196	0.0414	0.0043	0.1042	0.0058	0.0013	0.2291
5	−0.2	−0.185	0.0012	0.0065	−0.1932	0.0012	0.0060	−0.2016	0.0024	0.0121	−0.1933	0.0012	0.0060
6	−0.4	−0.404	0.0008	0.0019	−0.4036	0.0008	0.0019	−0.4124	0.0015	0.0036	−0.4036	0.0008	0.0019
7	−0.2	−0.205	0.0012	0.0058	−0.1992	0.0012	0.0061	−0.2025	0.0026	0.0127	−0.1993	0.0012	0.0061
8	0	0.028	0.0015	0.0553	−0.0058	0.0014	0.2402	−0.0063	0.0027	0.4338	−0.0061	0.0014	0.2299
9	0	0.032	0.0013	0.0416	−0.0030	0.0012	0.3932	−0.0235	0.0030	0.1271	−0.0031	0.0012	0.3779
10	0	0.046	0.0021	0.0452	−0.0060	0.0017	0.2859	−0.0019	0.0030	1.5957	−0.0062	0.0017	0.2760
11	0	0.015	0.0015	0.0943	−0.0079	0.0014	0.1722	0.0127	0.0040	0.3176	−0.0080	0.0014	0.1688
12	0	−0.001	0.0024	2.7776	−0.0121	0.0022	0.1850	0.0621	0.0065	0.1043	−0.0120	0.0022	0.1868

**Table 3 materials-15-08630-t003:** The results of shear frame: incomplete measurement case.

No.	*θ* _a_	FA			GA			PSO			m-NMFA		
		*θ*	SD	CV	*θ*	SD	CV	*θ*	SD	CV	*θ*	SD	CV
1	0	0.017	0.0024	0.0293	0.0045	0.0020	0.0459	0.0613	/	1.2185	−0.0026	0.0020	0.7461
2	0	0.0473	0.0015	0.0368	0.0383	0.0077	0.0150	0.1942	/	0.6339	0.0070	0.0044	0.6253
3	0	0.0436	/	0.0039	0.0036	0.0023	0.2470	0.1399	0.0023	0.0077	0.0199	0.0081	0.4085
4	0	0.0117	/	0.0244	0.0271	/	0.0141	0.1616	0.0097	0.0346	−0.0272	0.0062	0.2278
5	−0.2	−0.0973	0.0121	0.3250	−0.1704	0.0015	0.0026	0.1186	/	0.4796	−0.2036	0.0037	0.0184
6	−0.4	−0.3829	0.0038	0.0348	−0.3727	0.0069	0.0104	−0.3700	/	0.0330	−0.4072	0.0037	0.0091
7	−0.2	−0.2059	0.0037	0.0098	−0.0327	0.0018	0.0041	−0.0968	/	0.0248	−0.1977	0.0047	0.0235
8	0	0.0682	0.0036	0.1014	0.0266	0.0038	0.0056	0.1807	/	0.0366	−0.0156	0.0089	0.5712
9	0	−0.0001	0.0022	0.0083	0.0813	0.0048	0.0841	0.1135	/	0.4384	−0.0244	0.0071	0.2907
10	0	0.0468	0.0067	0.0263	−0.0314	0.0037	0.2940	0.0855	/	0.2556	0.0058	0.0062	1.0808
11	0	0.0299	0.0043	0.0431	0.0151	0.0045	0.0155	0.1274	0.0018	0.0393	0.0046	0.0027	0.5872
12	0	0.0211	0.0030	0.0867	0.0336	0.0018	0.0134	0.1086	/	0.0620	−0.0079	0.0030	0.3828

**Table 4 materials-15-08630-t004:** The first six frequencies of the shear frame identified by FDD (Unit: Hz).

Modal Order	Undamaged	Damaged
1	2.6616	2.6104
2	7.8329	7.3901
3	12.5790	12.3932
4	16.6569	16.5127
5	19.7584	19.0000
6	21.4556	21.2004

**Table 5 materials-15-08630-t005:** Model updating parameters of the undamaged structure using the Bayesian probability method.

*n*	1	2	3	4	5	6
${\hat{\theta }}_{n}^{\mathrm{ud}}$	0.4402	−0.0904	−0.1299	−0.1259	−0.0910	−0.0126
${\hat{\theta }}_{n}^{\mathrm{ud}}$	0.0051	0.0022	0.0033	0.0030	0.0028	0.0023

**Table 6 materials-15-08630-t006:** Model updating parameters of the damaged structure using the Bayesian probability method.

*n*	1	2	3	4	5	6
${\hat{\theta }}_{n}^{\mathrm{ud}}$	0.4923	−0.0792	−0.1438	−0.2008	−0.3099	−0.0250
${\hat{\theta }}_{n}^{\mathrm{ud}}$	0.0050	0.0022	0.0034	0.0032	0.0018	0.0023

## Data Availability

All data analyzed in this study have been included in this paper.

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
