# Peer review of "Probabilistic Structural Model Updating with Modal Flexibility Using a Modified Firefly Algorithm"

_materials, 2022, doi:10.3390/ma15238630_

Round 1
Reviewer 1 Report
Establish function by Bayesian probability method, the work is in nice shape and can be published in Materials after the following comments are properly addressed
1. Fig 2a needs more clear demonstration for cross-in-tray function.
2. Fig 2c needs scale down on interaction axis.
3. Fig 2g and h needs scale down on X aix.
4. Fig 2j needs scale up on Y axis.
5. Fig 4 a and b needs clear drawing on values of damage vector.
6. Fig 5 is not necessary in my opinion.
7. Fig 6 and 7 needs re-drawing.
8. Fig 8 needs standard deviation.
Overall, the work is well organised, however, some modification are necessary before its in-press.
Reviewer 2 Report
In this article, a Bayesian model updating method with modal flexibility was used to present a modified heuristic optimization algorithm named the modified Nelder-Mead firefly algorithm(m-NMFA); this model has the advantages of high precision, fast convergence, and strong robustness than other models. In my opinion, the paper can be published after making some minor revisions to the article, which are as follow:
1. As shown in Figure 2p, the x and y-axis are described in Chinese; it should be revised to English.
2. Figure 9-10 shows that the θ5 has more damage than other parameters; the author should describe the reason.
3. I suggest kindly enhancing the ratio of the latest references, for example:
i. Genetic-algorithm-based deep neural networks for highly efficient photonic device design [Photonics Research 9(6) B247-B252, (2021)],
ii. Ultrafast all-optical terahertz modulation based on an inverse-designed metasurface [Photonics Research 9(6) 1099-1108, (2021)],
iii. Enhanced photon communication through Bayesian estimation with an SNSPD array [Photonics Research 8(5) 637-641, (2020)].
Reviewer 3 Report
The article is very interesting, the introduction is properly described, the methodology is clearly presented. The article also raises an important topic from the point of view of readers' interest. Literature is also presented at an appropriate level. However, according to the reviewer, this is not the right magazine, but I leave that decision to the publisher - in my opinion, it should be a magazine devoted to the simulation of construction or, ultimately, the strength of materials.
